# MEKK-3 Acts Cooperatively with NSY-1 in SKN-1-Dependent Manner against Oxidative Stress and Aging in *Caenorhabditis elegans*

**DOI:** 10.3390/biology11101526

**Published:** 2022-10-18

**Authors:** Min Hwang, Chandani Shrestha, Shinwon Kang, Jiyoon Kim

**Affiliations:** 1Department of Pharmacology, Department of Biomedicine & Health Sciences, College of Medicine, The Catholic University of Korea, Seoul 06591, Korea; 2Department of Physiology, University of Toronto, Toronto, ON M5S, Canada; 3Lunenfeld-Tanenbaum Research Institute, Mount Sinai Hospital, Sinai Health System, Toronto, ON M5G, Canada

**Keywords:** longevity, oxidative stress, *Caenorhabditis elegans*, MEKK-3, NSY-1, SKN-1

## Abstract

**Simple Summary:**

Oxidative stress has been linked to myriad pathologies and aging. Here, we show that MEKK-3- and NSY-1-mediated nuclear translocation of SKN-1 is critical for protection against oxidative stress and for longevity in *Caenorhabditis elegans*. A genetic approach revealed that the knockdown of MEKK-3 and NSY-1, components of the p38 MAPK pathway, significantly reduced the resistance to oxidative stress achieved by SKN-1 overexpression. Mechanistic analyses showed that MEKK-3 and NSY-1 participate in the accumulation of SKN-1 in intestinal nuclei, which is required for the activation of phase II detoxification genes. These results provide molecular insights into oxidative stress resistance and reveal potential targets for aging-associated diseases.

**Abstract:**

Oxidative stress resulting from reactive oxygen species and other toxic metabolites is involved in human diseases, and it plays an important role in aging. In *Caenorhabditis elegans*, SKN-1 is required for protection against oxidative stress and aging. As p38 mitogen-activated protein kinase signaling is activated in response to oxidative stress, SKN-1 accumulates in intestinal nuclei and induces phase II detoxification genes. However, NSY-1, a well-known mitogen-activated protein kinase kinase kinase (MAPKKK) of *C. elegans*, acts as a partial regulator of the SKN-1-induced oxidative stress signaling pathway, suggesting that the regulator for optimal activation of SKN-1 remains unknown. Here, we report a MAPKKK, MEKK-3, as a new regulator required for full activation of SKN-1-mediated resistance against oxidative stress and aging. In RNA-interference-based screening, we found that the simultaneous knockdown of *mekk-3* and *nsy-1* significantly decreased the oxidative stress resistance and survival of SKN-1 transgenic worms. MEKK-3 was induced in response to oxidative stress. Mechanistic analysis revealed that double knockdown of *mekk-3* and *nsy-1* completely suppressed the nuclear localization of SKN-1. These results were reproduced in mutant worms in which SKN-1 is constitutively localized to intestinal nuclei. In addition, *mekk-3* and *nsy-1* were required for optimal induction of SKN-1 target genes such as *gcs-1* and *trx-1*. These data indicate that MEKK-3 plays an essential role in the SKN-1-dependent signaling pathway involved in oxidative stress resistance and longevity by cooperating with NSY-1.

## 1. Introduction

Aging is a complex process involving natural degradation [1]. Several theories have focused on the mechanisms of aging, including free radicals, metabolic stress, and telomere dysfunction [2,3,4]. According to the free radical aging theory, oxidative damage to cells and tissues causes aging, and this mechanism has been implicated in the pathophysiology of numerous diseases [5,6]. It has been hypothesized that aging is caused by a combination of increased generation of reactive oxygen species (ROS) and decreased cellular antioxidant defenses, leading to oxidative stress [7].

*Caenorhabditis elegans* (*C. elegans*) is an ideal model system for studying oxidative stress and aging. The application of this model has a number of benefits, which include a short lifespan, rapid generation, simple cultivation, and well-established genetic pathways [8]. In *C. elegans*, the oxidative stress response is orchestrated by the transcription factor SKN-1, the ortholog of mammalian Nrf protein [9]. SKN-1 is required to regulate proteostasis and longevity [9,10,11]. In addition, SKN-1 has been reported to be regulated by the p38 mitogen-activated protein kinase (MAPK) pathway [12]. These positive and negative regulators of intestinal SKN-1 are affected by oxidative stresses, resulting in enhanced nuclear localization and transcriptional activation, thereby maintaining redox homeostasis. SKN-1 initiates the development of the feeding and digestive systems during the earliest embryonic stages [13,14,15]. While SKN-1 constitutively stimulates phase II gene expression in ASI neurons [16], oxidative stress is necessary to promote SKN-1 accumulation in the intestinal nucleus and induce gene expression.

Stress-activated protein kinases, which include the p38 MAPKs, are a subfamily of MAPKs that has been conserved throughout evolution and play a significant role in the processes of adaptation, homeostasis, and specialized stress responses. The p38 MAPK signaling pathway is involved in the cellular response to a variety of stressors and inflammatory cytokines, according to research conducted in cell culture systems derived from vertebrates [17]. However, relatively insufficient research has been conducted on the role of the p38 signaling pathway in the stress response. 

The *C. elegans* p38 MAPK signaling pathway is regulated by PMK-1 MAPK (the ortholog of mammalian MAPK11 and MAPK14), SEK-1 MAPK kinase (MAPKK, the ortholog of mammalian MAP2K6), and NSY-1 MAPKK kinase (MAPKKK, the ortholog of mammalian MAP3K5 and MAP3K15), which function in the intestine to phosphorylate SKN-1, leading to its stabilization and nuclear translocation [17,18,19]. However, under oxidative stress conditions, NSY-1 acts as a partial regulator of the oxidative stress signaling pathway, which is mediated by SKN-1 [20], suggesting that an additional pathway or regulator molecule activates SKN-1 via interaction with PMK-1. 

In this study, we evaluated the genetic mechanism underlying the expression of SKN-1, which plays a role in the oxidative stress response. For this purpose, we performed RNA-interference-based screening of novel MAPKKKs that regulate SKN-1 under oxidative stress conditions in *C. elegans* as a model for stress-induced signal transduction. We found that double knockdown of *nsy-1* and *mekk-3* significantly decreased the survival rate and oxidative stress resistance of *C. elegans* and downregulated the expression of SKN-1 target genes, such as *gcs-1/glutamate-cysteine ligase* and *trx-1/thioredoxin-1*, compared to single knockdown. Our data indicate that MEKK-3 plays a role in oxidative stress-induced activation of the SKN-1 and p38 MAPK pathways in *C. elegans*.

## 2. Materials and Methods

### 2.1. Strain Maintenance

The following strains were maintained on nematode growth medium (NGM) and *Escherichia coli* OP50 at 20 °C using standard methods [21]: *N2* Bristol as wild-type, *N2* Ex001[SKN-1::GFP], *N2* Ex020[SKN-1 S393A::GFP], *gcs-1::gfp*, *trx-1::gfp*, and *mekk-3::gfp*. All strains of *C. elegans* were obtained from the Caenorhabditis Genetics Center (Minneapolis, MN, USA) unless otherwise specified.

### 2.2. Plasmid Construction

The MEKK-3::GFP translational fusion was created by amplifying and cloning the *mekk-3* cDNA from the wild-type cDNA pool into the GFP vector pPD95.67. The MEKK-3 genomic region was cloned directly behind the inducible lac promoter in a frame containing GFP and controlled by oxidative stress. All PCRs were performed using Pfu polymerase (Merck Millipore, Burlington, MA, USA). All genetic information was obtained from http://www.wormbase.org (Release WS284) accessed on 1 April 2020. Young adult *N2* animals were injected with a DNA mixture containing the *mekk-3::gfp* fusion construct (50 ng/μL) and pRF4rol-6 (100 ng/μL) as markers. Three F1 roller worms were placed on a plate and enabled to reproduce by themselves. RNA interference (RNAi) clones were constructed by PCR amplification of full-length *mekk-3*, *nsy-1*, *dlk-1*, *tap-1*, *mom-4*, *zak-1*, *irk-1*, *mtk-1*, and *kin-18* genes. RNAi clones were obtained from Horizon Discovery (Cambridge, U.K.). 

### 2.3. RNAi

RNAi experiments were performed by feeding the worms as previously described, with minor modifications [22]. Plates for RNAi were made by adding ampicillin at a concentration of 100 g/mL and isopropyl β-D-1-thiogalactopyranoside (IPTG, Merck-Millipore, Burlington, MA, USA) at a concentration of 2 mM to NGM. RNAi clones were grown in LB medium that was treated with 100 µg/mL ampicillin and 12.5 µg/mL tetracycline at 37 °C. The following day, the cultures were diluted (1:50), grown to an OD_600_ of 1, and induced with 1 mM IPTG. This culture was used for seeding plates containing tetracycline, ampicillin, and IPTG and left to dry for 1–2 days. The gene knockdown experiments were performed by transferring the worms on these plates, including RNAi bacteria.

### 2.4. GFP Fusion Protein Scoring System

SKN-1 accumulation in the intestinal nuclei was scored as previously described [16]. Strong nuclear SKN-1::GFP was observed throughout the entire intestine (“High”), while high levels of SKN-1::GFP were observed anteriorly or posteriorly but were barely detectable through the intestine (“Medium”). Fluorescence images were acquired using an Axiovert 200 microscope (Zeiss, Germany) and measured using the ImageJ software (National Institutes of Health, Bethesda, MD, USA).

### 2.5. Oxidative Stress Resistance Assay

To investigate resistance against oxidative stress, young adult worms were transferred onto plates containing 7.5 or 20 mM tert-butyl hydrogen peroxide (t-BOOH, Merck Millipore) in NGM agar at 20 °C. Worms were considered dead when they stopped responding to repeated pick prodding.

### 2.6. Lifespan Assay

Animals were synchronized by laying timed eggs overnight and were allowed to develop at 20 °C. When animals reached L4 or young adulthood, they were transferred to plates with OP50, RNAi bacteria, or t-BOOH; kept at 20 °C, and periodically scored as alive or dead in response to prodding. The data did not include nematodes or worms that escaped or died of vulval bursting.

### 2.7. RNA Isolation and Quantitative Real-Time PCR (qRT-PCR)

Mixed-stage worms were collected from each strain and then washed three times with M9 buffer. Total RNA was isolated using TRI Reagent (Merck Millipore), according to the manufacturer’s protocol. The concentration and purity of RNA were determined using a NanoDrop 2000 spectrophotometer (Thermo Fisher, Waltham, MA, USA). cDNA was synthesized using 2.5 µg of RNA and the SuperScript III cDNA synthesis kit (Merck-Millipore), according to the manufacturer’s specifications. Gene expression levels were determined by qRT-PCR using SYBR Premix Ex Taq II (Takara Biotechnology, Dalian, China) and ViiA7 System (Thermo Fisher). The amount of target mRNA was normalized to that of act-1 mRNA. The primer sequences used are listed in Appendix A.

### 2.8. Statistics

Statistical analyses were performed using Prism 9 (GraphPad Software, Inc., La Jolla, CA, USA). Statistical significance was determined using Student’s *t*-test for comparing two means or a one-way/two-way analysis of variance with a post hoc test (Dunnett, Tukey) for comparing multiple means. Statistical significance was set at *p* < 0.05. Statistical analyses for survival were conducted using Log-rank (Mantel-Cox) test. Statistical analyses for SKN-1 nuclear localization were conducted using a chi^2^ test. Data are expressed as the mean ± SEM.

## 3. Results

### 3.1. Concomitant Inhibition of mekk-3 and nsy-1 Significantly Suppresses the Oxidative Stress Resistance of SKN-1 Transgenic Worms

To identify a regulator of SKN-1 in the p38 MAPK pathway in *C. elegans*, we screened novel MAPKKKs that regulate SKN-1 under oxidative stress conditions by comparing the survival rates of SKN-1 transgenic worms treated with RNAi against the candidate genes. Since NSY-1 is a partial regulator of SKN-1 in the oxidative stress pathway [20], we screened by simultaneous inhibition of *nsy-1* and MAPKKKs, including *dlk-1* (the ortholog of mammalian MAP3K12)*, tap-1* (the ortholog of mammalian TAB1)*, mom-4* (the ortholog of mammalian MAP3K7)*, zak-1* (the ortholog of mammalian MAP3K20)*, irk-1* (the ortholog of mammalian KCNJ14, KCNJ16, and KCNJ18)*, mtk-1* (the ortholog of mammalian MAP3K4)*, kin-18* (the ortholog of mammalian TAOK1, TAOK2, and TAOK3)*,* and *mekk-3* (the ortholog of mammalian MAP3K3) using specific RNAi (Figure 1a). The knockdown efficiency of each RNAi was confirmed by qRT-PCR (Appendix A). SKN-1 transgenic worms displayed an extraordinarily high survival rate even when exposed to 5–10 mM t-BOOH, an exogenous inducer of ROS, in the conventional oxidative stress assay (Appendix A); therefore, an extraordinarily high concentration of 20 mM t-BOOH was used for the screening. We found that concomitant inhibition of *mekk-3* and *nsy-1* showed the most significant decrease in the RNAi screening (Figure 1a) and completely suppressed resistance against oxidative stress compared to a single knockdown (Figure 1b). These results showed that *mekk-3* RNAi suppressed the oxidative stress resistance achieved through the overexpression of SKN-1, and *mekk-3* partially complements the function of *nsy-1*.

### 3.2. MEKK-3 Overexpression Displays Increased Resistance against Oxidative Stress

To determine the expression pattern of MEKK-3, we analyzed the expression of a transgene in which GFP was fused to the C-terminus of full-length MEKK-3 (MEKK-3::GFP). Expression of MEKK-3::GFP was observed in the intestine of wild-type (*N2*) young adult worms (Figure 2a). Notably, after treatment with 7.5 mM t-BOOH at 20 °C for 16 h, the intestinal cells showed high levels of MEKK-3::GFP (Figure 2a,b), and the mRNA expression of MEKK-3 was significantly increased (Figure 2c). In addition, we tested whether MEKK-3 overexpression increased resistance against oxidative stress. Surprisingly, MEKK-3:: GFP transgenic worms showed approximately three times higher resistance to oxidative stress than *N2* (Figure 2d and Appendix A); however, their longevity was not significantly higher than that of *N2* (Appendix A). Collectively, these results suggest that MEKK-3 overexpression has a strong positive effect on resistance against oxidative stress; however, it is not sufficient to extend the lifespan.

### 3.3. MEKK-3 Is Required for the Nuclear Localization of SKN-1 

Oxidative stress activates the p38 MAPK pathway [23]. In p38 kinase signaling in *C. elegans*, NSY-1 phosphorylates SEK-1, which phosphorylates PMK-1 and is required for SKN-1 accumulation in the intestinal nuclei during oxidative stress [12]. Because MEKK-3 was significantly induced by oxidative stress (Figure 2a–c), we investigated whether the oxidative stress-induced translocation of SKN-1 into the nucleus requires MEKK-3. RNAi-mediated knockdown of *mekk-3* in SKN-1 transgenic worms significantly decreased the nuclear localization of SKN-1 under normal and oxidative stress conditions (Figure 3a,b; *mekk-3* RNAi). Notably, simultaneous knockdown of *mekk-3* and *nsy-1* completely suppressed the nuclear translocation of SKN-1 (Figure 3a,b; *mekk-3*&*nsy-1* RNAi). To investigate whether *mekk-3* is required for the longevity of SKN-1 transgenic worms, we examined the lifespan. Interestingly, while the knockdown of *mekk-3* or *nsy-1* did not significantly change the lifespan, double knockdown of *mekk-3* and *nsy-1* decreased the lifespan of SKN-1 transgenic worms by 25% (Figure 3c, Appendix A).

Next, we investigated the effects of MEKK-3 and NSY-1 on SKN-1 localization using a phosphorylation-inhibiting mutant in which serine residues at 393 of SKN-1::GFP were replaced with alanine (SKN-1 S393A::GFP). In a previous study, the SKN-1 S393A::GFP mutant was constitutively localized to the intestinal nuclei by alteration of the critical inhibitory glycogen synthase kinase-3 (GSK-3) phosphorylation site; it increased oxidative stress resistance compared to *N2* SKN-1::GFP [24]. We found that the nuclear localization of SKN-1 S393A::GFP was strong in the intestinal nuclei but was significantly decreased by genetic knockdown of *mekk-3* and *nsy-1* via RNAi (Figure 3d,e). These results also exclude the possibility that *sek-1* and p38 signaling is simply epistatic to *gsk-3* with respect to SKN-1 regulation. Interestingly, the individual and synergistic effects of *mekk-3* and *nsy-1* knockdown were more pronounced on the lifespan of SKN-1 S393A transgenic worms than on SKN-1 transgenic worms (Figure 3f, Appendix A). This is possibly because SKN-1-dependent lifespan extension in the absence of GSK-3 inhibitory phosphorylation enhances the effect of *mekk-3* and *nsy-1* knockdown on longevity. Collectively, these results demonstrated that MEKK-3 is required for the nuclear translocation of SKN-1 and that it plays a synergistic role with NSY-1.

### 3.4. MEKK-3 Is Required for Optimal Activation of SKN-1 Target Genes

SKN-1 promotes the expression of phase II detoxification genes, which encode enzymes responsible for antioxidants and glutathione synthesis [13,16]. The significant role of MEKK-3 was determined via examination of SKN-1 target genes, such as *gcs-1* and *trx-1*, which encode the gamma-glutamine cysteine synthase heavy chain and thioredoxin, respectively, and was induced by oxidative stress (Appendix A). To investigate whether MEKK-3 regulates SKN-1 target genes, we used transgenic worms overexpressing *gcs-1* with C-terminal GFP (GCS-1::GFP) or *trx-1* with C-terminal GFP (TRX-1::GFP). Under oxidative stress conditions, *mekk-3* and *nsy-1* knockdown significantly decreased the expression of GCS-1::GFP (Figure 4a,b) and mRNA expression of *gcs-1* (Figure 4c). Similarly, the expression of TRX-1::GFP (Figure 4d,e) and mRNA expression of *trx-1* (Figure 4f) were significantly decreased following *mekk-3* and *nsy-1* knockdown. These results indicate that MEKK-3 is a key factor in the fine regulation and optimal activation of detoxification genes induced by SKN-1.

## 4. Discussion

In the present study, we showed that MEKK-3 is critically involved in SKN-1-dependent resistance against oxidative stress and aging, particularly by regulating the translocation of SKN-1 into the nucleus and its target genes. We found that MEKK-3 overexpression acts as a sensor of oxidative stress (Figure 2a–c). Notably, MEKK-3 exhibited a synergistic effect with NSY-1 in response to oxidative stress (Figure 1, Figure 3 and Figure 4). Previous reports have suggested that NSY-1 is a partial regulator of SKN-1 [20], presumably through another regulatory factor that is redundant with NSY-1. The authors hypothesized that an additional pathway would be required to activate SKN-1 via interaction with SEK-1 or PMK-1. However, the results of the present study, surprisingly, showed that MEKK-3, another MAPKKK of the p38 MAPK pathway, could partially supplement the role of NSY-1, implying that the evolutionary adaptation of existing regulators with similar functions is more efficient than the development of new molecular pathways.

The dual specificity of kinases in the MAPK pathway has been reported by multiple pieces of literature [25,26]. For example, MEK-1 and MEK-2 are dual-specificity kinases that phosphorylate both ERK substrates (ERK1 and ERK2) with high selectivity [27]. Notably, single phosphorylation of either threonine or tyrosine residues leads to partial activation of ERK protein, while dual phosphorylation of both residues is needed for complete activation of ERK. In addition, MEK proteins require dual phosphorylation of two serine residues by several kinases, including Raf, TPL2, and Mos [28]. Although the universality of the principle of dual phosphorylation in kinase-mediated pathways is unclear, dual phosphorylation mediated by kinases and phosphatases can provide a more precise regulatory system favorable to the cellular defense systems and the survival of the organism.

Recently, MEKK-3 was reported to be a regulator of the nutrient-sensing pathway and an initiator of the diet-restriction (DR) response [29] as MEKK-3-deficient *C. elegans* phenocopies a DR-like state. The authors suggested that knocking down *mekk-3* induces metabolic reprogramming, where β-oxidation is upregulated, and production of ROS is reduced, while xenobiotic detoxification genes such as *pha-4*, *nhr-8*, and *ahr-1*/*aha-1* are upregulated. In addition, they suggested that the coupling of the metabolism shift and xenobiotic detoxification leads to extended longevity during *mekk-3* knockdown-mediated DR. Interestingly, the effect of *mekk-3* knockdown on lifespan extension depends on temporal conditions of the onset of RNAi onset. Knocking down the *mekk-3* gene at L4 or later produced no effect, while the maximum extension of lifespan was observed when *mekk-3* RNAi was initiated at the L1 stage [29], suggesting that the temporal requirements of *mekk-3* are different from those of the insulin/IGF-1 signaling pathway and are similar to those of mitochondrial genes that affect lifespan [30]. Therefore, the results of the present study are consistent with those of previous reports (Figure 3c,f). Further investigation of *mekk-3* requirements, including the potential role of *nsy-1* in this condition, may contribute to the elucidation of the mechanism against cellular stresses such as metabolic or oxidative stress.

## 5. Conclusions

We used genetic and mechanistic approaches to demonstrate that MEKK-3 plays a key physiological role in the SKN-1-mediated oxidative stress pathway. MEKK-3 and NSY-1 deficiencies lead to defects in the nuclear localization of SKN-1, which decreases resistance against oxidative stress and aging due to a reduction in the expression of SKN-1 target genes such as *gcs-1* and *trx-1*. In addition, high levels of MEKK-3 induced by oxidative stress would further increase the resistance against oxidative stress.

Human Nrf protein-mediated ROS protection is thought to be helpful in a wide variety of diseases, including diabetes, neurodegenerative diseases, atherosclerosis, and viral infection. This gene activation may offer a generally applicable method of cancer prevention by aiding in drug detoxification, which is crucial for chemotherapy tolerance. For example, the consumption of chemoprotective antioxidants, which are mediated by Nrf2, prevents the development of chemical carcinogenesis in mice [31] and lowers the incidence of gastrointestinal and lung cancer in humans [32,33]. The functional similarities between SKN-1 and Nrf proteins would open the way for a new therapeutic by exploring these SKN-1-mediated pathways in *C. elegans*, which is particularly appropriate for genetic and pharmacological screening.

Increasing evidence also suggests that the p38 MAPK pathway plays a pivotal role in aging-related diseases, such as cancer, arthritis, diabetes, and neurodegenerative diseases [34,35]. Therefore, the finding that MEKK-3 deficiency resulted in reduced resistance to oxidative stress and aging has enormous implications in the search for druggable targets against aging-associated diseases, as well as in elucidating the physiological mechanisms of aging.

## Figures and Tables

**Figure 1 biology-11-01526-f001:**
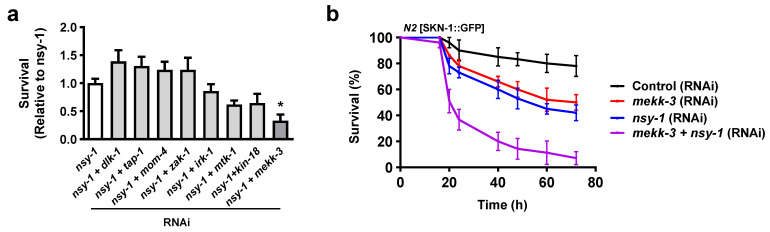
Genetic screening of a novel MAPKKK in SKN-1 transgenic *C. elegans*. (**a**) RNAi screening was performed in SKN-1 transgenic worms to identify MAPKKKs involved in the resistance to oxidative stress. Worms were fed with bacteria for each RNAi treatment 24 h before exposure to oxidative stress. Oxidative stress resistance assays were performed 16 h after RNAi, and the survival rate was calculated (*n* = 3). Data are shown as the mean ± SEM. * *p* < 0.05. All *p* values were calculated by ANOVA followed by Dunnett’s multiple comparison tests. (**b**) Oxidative stress resistance assay was performed on SKN-1 transgenic worms fed RNAi bacteria against *mekk-3*, *nsy-1*, or both *mekk-3* and *nsy-1* under treatment with 20 mM t-BOOH. Survival rate was calculated at the indicated time points (h). This representative experiment involved 30 worms in each group. Data are shown as the mean ± SEM.

**Figure 2 biology-11-01526-f002:**
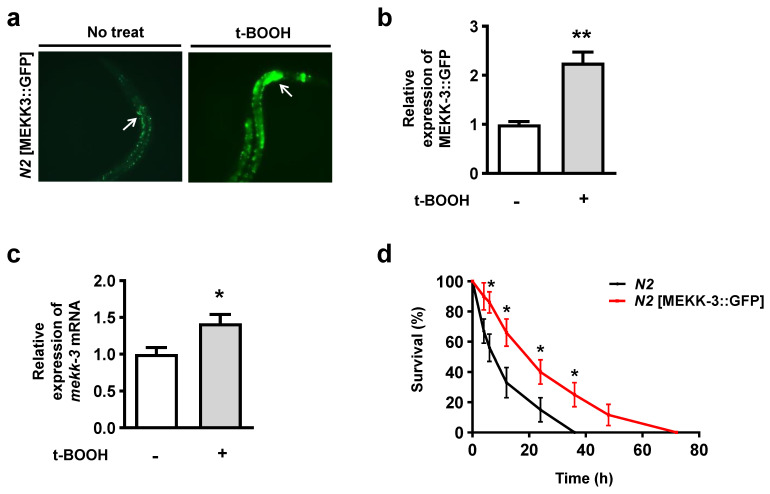
Expression of MEKK−3 in oxidative stress. (**a**) Localization of MEKK-3 labeled with GFP under normal conditions (left) and under treatment with 7.5 mM t-BOOH for 24 h (right). MEKK-3 transgenic worms were visualized by GFP expression using fluorescence microscopy. The white arrows indicate the MEKK-3::GFP protein in the intestine. (**b**) Quantification of MEKK-3::GFP was performed using the ImageJ software and summarized (*n* = 3). (**c**) Quantitative PCR analyses of *mekk-3* (*n* = 3). Total RNA extraction from MEKK-3 transgenic worms was performed 24 h after the exposure to 7.5 mM t-BOOH. (**d**) Oxidative stress resistance assay for MEKK-3 transgenic worms. The survival rate was calculated at the indicated time points (h) after the exposure to 7.5 mM t-BOOH. This representative experiment involved 30 worms in each group. Data are shown as the mean ± SEM. * *p* < 0.05. ** *p* < 0.01. All *p* values were calculated by unpaired two-tailed Student’s *t*-tests.

**Figure 3 biology-11-01526-f003:**
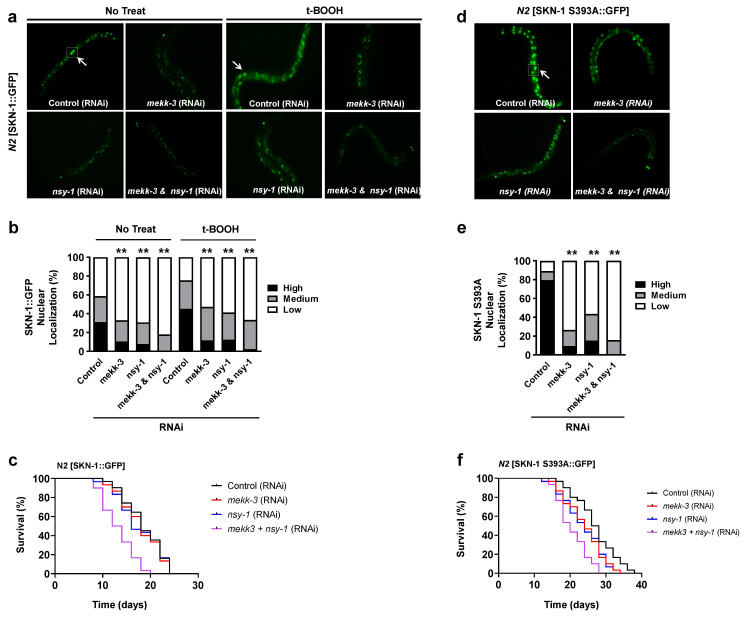
Knockdown of MEKK−3 and NSY−1 completely suppresses the nuclear localization of SKN−1 and SKN−1−dependent oxidative stress resistance and longevity. (**a**) The nuclear localization of SKN-1 labeled with GFP in the intestine. SKN-1 transgenic young adult worms were placed on NGM plates that contained either *mekk-3*, *nsy-1*, or a combination of *mekk-3* and *nsy-1* RNAi bacteria for 24 h. The L4440 vector without insertion in bacteria was used as a control. Fluorescent images were captured 4 h after the transfer of worms fed RNAi bacteria on the normal (left) or 20 mM t-BOOH (right) plates. The white arrows indicate SKN-1::GFP. (**b**) Young adult worms were scored for accumulation of SKN-1::GFP in the intestinal nuclei (n = 30 in each group). (**c**) Lifespan assay was performed in SKN-1 transgenic worms. Young adult worms were transferred to plates with bacteria containing RNAi plasmids against *mekk-3*, *nsy-1*, or both *mekk-3* and *nsy-1*. The L4440 vector without insertion in bacteria was used as a control. The percentage of live animals was plotted at the indicated time points (days). This experiment involved 90 worms in each group. The mean and median lifespan are shown in Appendix A. (**d**) The nuclear localization of SKN-1 S393A mutant labeled with GFP. All procedures were performed in the same manner as described in (**a**). SKN-1 nuclear localization of each inset was confirmed by using DAPI staining in Appendix A. (**e**) Quantitative analyses of the fluorescent images of SKN-1 S393A::GFP. All procedures were performed in the same manner as described in (**b**) (n = 30 in each group). (**f**) Lifespan assay was performed in SKN-1 S393A transgenic worms. All procedures were performed in the same manner as described in Figure 1b. This experiment involved 90 worms in each group. The mean and median lifespan are shown in Appendix A. ** *p* < 0.01 compared to control. All *p* values were calculated by chi^2^ tests.

**Figure 4 biology-11-01526-f004:**
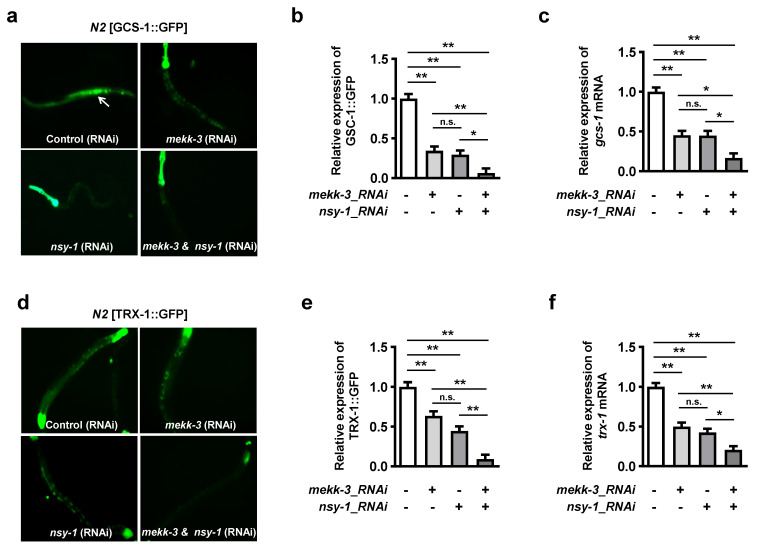
MEKK−3 and NSY−1 are indispensable for the induction of SKN−1 target genes. (**a**) Intestinal expression of GCS-1 labeled with GFP. GCS-1 transgenic young adult worms were fed bacteria that carry RNAi targeting either *mekk-3*, *nsy-1*, or both *mekk-3* and *nsy-1* RNAi for 24 h. The L4440 vector without insertion in bacteria was used as a control. Fluorescent images were captured 6 h after the transfer of worms fed RNAi bacteria on 7.5 mM t-BOOH plates. The white arrows indicate GCS-1::GFP. (**b**) Intestinal GCS-1::GFP expression was scored in young adult worms (n = 30 in each group). (**c**) Quantitative PCR analyses of *gcs-1* (n = 3). Total RNA extraction from GCS-1 transgenic worms was performed 6 h after the exposure to 7.5 mM t-BOOH. (**d**) Intestinal localization of TRX-1 labeled with GFP. All procedures were performed as described in (**a**). (**e**) Quantitative analyses of the fluorescent images of TRX-1::GFP. All procedures were performed as described in (**c**) (n = 30 for each group). (**f**) Quantitative PCR analyses of *trx-1* (n = 3). All procedures were performed as described in (**c**). Data are shown as the mean ± SEM. n.s.: not significant, * *p* < 0.05, ** *p* < 0.01. All *p* values were calculated by two-way ANOVA followed by Tukey’s multiple comparison tests.

## Data Availability

Not applicable.

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
