# Peer review of "MEKK-3 Acts Cooperatively with NSY-1 in SKN-1-Dependent Manner against Oxidative Stress and Aging in Caenorhabditis elegans"

_biology, 2022, doi:10.3390/biology11101526_

Round 1
Reviewer 1 Report
This manuscript describes the role of MEKK-3 and NSY-1 in SKN-1-depedent responses including lifespan in C. elegans. Overall, the topic is important in the field and experiments were well designed and manuscript is well written.
Please include names of protein/genes when it is first used and/or names of homologs for easy understanding.
Results of lifespan (or survival) should include mean and median lifespans as well as sample size from each experiment. Not sure how many worms were used for each exp. in Fig. 2d. Also it is not clear why survival data in fig. 3 are different from earlier lifespan data. In fig. 3c and 3F, it is not clear if sample size is 30 or 90.
Please include standard error bars for controls in Fig. 1,2, and 4.
It is not clear what statistical method was used to compare results in Fig. 3B and 3E.
Results in Fig. 4 should be analyzed by 2-way ANOVA to determine the interaction of 2 variables tested. These results should also compare between groups other than the control.
Author Response
Please see the attachment. We have made every attempt to carefully and thoroughly address your suggestions and concerns. We have included our point-by-point responses to the concerns herein. We believe that our manuscript has improved substantially based on these changes.

Reviewer 2 Report
The manuscript by Kim et al entitled "MEKK-3 acts cooperatively with NSY-1 in SKN-1-dependent 2 manner against oxidative stress and aging in Caenorhabditis elegans" describes the regulation of MEKK-3 and NSY-1 cooperate in SKN1-dependent manner against oxidative stress”.
In this manuscript, the author Unravelling the role of MEKK-3 and NSY-1 regulation SKN-1 in the presence of oxidative stress.
One of the main problems with this manuscript is that the author uses transgenic C. elegans in the majority of the experiments, and that transgene is controlled by a constitutive promoter. Under oxidative stress, transgenic expression is controlled not by changes in promoter activity but rather by protein accumulation or breakdown.
In experiment 3.3 author demonstrates that MEKK-3 requires for nuclear localization of SKN-1: in this experiment, the author should show the DAPI staining to confirm the nuclear staining.
SKN-1 S393A strain shows the constitutive nuclear localization of SKN-1 and MEKK-3 or NSY-1 is an upstream protein for regulation of SKN1 protein activity. How it is possible that constitutive active SKN-1 will be affected with MEKK-3 and NSY-1 under normal conditions or under oxidative stress.
Provide the protocol for the knockdown of the gene by siRNA.
Author Response

(The authors gave the same response as above.)

Round 2
Reviewer 2 Report
the manuscript is well-written and each experiment is explained well.